# Short report: Plasma based biomarkers detect radiation induced brain injury in cancer patients treated for brain metastasis: A pilot study

Chen Makranz[1], Asael Lubotzky[2,3], Hai Zemmour[2], Ruth Shemer[2], Benjamin Glaser[2], Jonathan Cohen[4,5], Myriam Maoz[4], Eli Sapir[6,7], Marc Wygoda[6], Tamar Peretz[4], Noam Weizman[6], Jon Feldman[6], Ross A. Abrams[6], Alexander Lossos[1], Yuval Dor[2], Aviad Zick[4]*

1 Department of Neurology and Oncology, The Gaffin Center for Neurooncology, Sharett Institute for Oncology, Hadassah Medical Center, Faculty of Medicine, Hebrew University of Jerusalem, Jerusalem, Israel, 2 Department of Developmental Biology and Cancer Research, Institute for Medical Research Israel-Canada, the Hebrew University-Hadassah Medical School, Jerusalem, Israel, 3 Division of Neurology and Department of Molecular Genetics, The Hospital for Sick Children, University of Toronto, Toronto, Ontario, Canada, 4 Department of Oncology, Sharett Institute for Oncology, Hadassah Medical Center, Faculty of Medicine, Hebrew University of Jerusalem, Jerusalem, Israel, 5 The Wohl Institute for Translational Medicine, Hadassah Medical Center, Jerusalem, Israel, 6 Department of Radiation Oncology, Sharett Institute for Oncology, Hadassah Medical Center, Faculty of Medicine, Hebrew University of Jerusalem, Jerusalem, Israel, 7 Radiation Oncology Institute, Samson Assuta Ashdod University Hospital, Ben Gurion University, Ashdod, Israel

* aviadz@hadassah.org.il

**Data Availability Statement:** All relevant data are within the manuscript and its Supporting Information files.

## Abstract

### Background

Radiotherapy has an important role in the treatment of brain metastases but carries risk of short and/or long-term toxicity, termed radiation-induced brain injury (RBI). As the diagnosis of RBI is crucial for correct patient management, there is an unmet need for reliable biomarkers for RBI. The aim of this proof-of concept study is to determine the utility of brain-derived circulating free DNA (BncfDNA), identified by specific methylation patterns for neurons, astrocytes, and oligodendrocytes, as biomarkers brain injury induced by radiotherapy.

### Methods

Twenty-four patients with brain metastases were monitored clinically and radiologically before, during and after brain radiotherapy, and blood for BncfDNA analysis (98 samples) was concurrently collected. Sixteen patients were treated with whole brain radiotherapy and eight patients with stereotactic radiosurgery.

### Results

During follow-up nine RBI events were detected, and all correlated with significant increase in BncfDNA levels compared to baseline. Additionally, resolution of RBI correlated with a decrease in BncfDNA. Changes in BncfDNA were independent of tumor response.

**Funding:** This research is funded by Israel Cancer Research Foundation 2016 (to AZ), The Katz Memorial Fund 2017 (to CM), Prusiner-Abramsky Research Award in clinical and basic Neuroscience 2019 (to CM), Esther Monsa Fund 2018 (to CM), Joint Research Fund of the Hebrew University of Jerusalem and Hadassah Medical Center 2016 (to YC). This work is also supported in part by grants of the Ernest and Bonnie Beutler Research Program of Excellence in Ge-nomic Medicine, The Israel Science Foundation, The Waldholtz/Pakula family, The Robert M. and Marilyn Stern-berg Family Charitable Foundation (to YD). Israel Cancer Research Foundation and the Rothschild Fellowship for Physician- Researchers (AsL) The funders had no role in study design, data collection and analysis, decision to publish, or preparation of the manuscript.

**Competing interests:** The authors have declared that no competing interests exist.

**Abbreviations:** BBB, Blood-brain-barrier.; BncfDNA, brain-derived circulating free DNA.; cfDNA, Circulating free DNA.; FC, fold change.; GBM, Glioblastoma multiforme.; log2, logarithmic scale.; PD, Progressive Disease.; TR, Tumor response.; TRAM, Treatment response assessment maps.; RBI, Radiation induced brain injury.; SD, stable disease.; SRS, Stereotactic radiosurgery.; URP, Undetermined Radiological Progression.; WBRT, whole brain radiotherapy..

## Conclusions

Elevated BncfDNA levels reflects brain cell injury incurred by radiotherapy. further research is needed to establish BncfDNA as a novel plasma-based biomarker for brain injury induced by radiotherapy.

## Background

Brain radiotherapy plays an important role in the management of brain metastasis [1–3]. Although effective, brain radiotherapy may cause both short and/or long-term toxicity leading to reduction in quality of life[3, 4]. This toxicity, referred as radiotherapy-induced brain injury (RBI) may occur during and/or after treatment and is classified as acute (within days and up to 4 weeks after the end of radiotherapy), early-delayed (1 month to 6 months after radiotherapy), or late-delayed (later than 6 months to several years after treatment) [5–7]. The pathophysiol-ogy of RBI is not well understood. However, evidence supports a multifactorial process leading to apoptosis and functional alternations across multiple brain cell types, and vascular changes leading to disruption of the blood-brain-barrier (BBB) [6, 8–11].

Currently, diagnosis of RBI is clinical and radiologic, and may be difficult to distinguish from tumor progression [12–18]. While early onset RBI, is mostly reversible, late appearing RBI, which can present either as radio-necrosis or as periventricular leukoencephalopathy, is mostly irreversible [5, 6, 9, 18–20]. Given the critical importance of accurate diagnosis of RBI for optimal patient management, there is an unmet need for sensitive, specific, and readily available diagnostic tools to monitor RBI development. A validated, reliable set of biomarkers to detect and monitor RBI would be one way of meeting this challenge. In this study we aimed to explore the utility of plasma-based biomarkers for detection and monitoring RBI using a previously published approach [21–26] based on identifying cell-specific methylation patterns in circulating free DNA (cfDNA). Since DNA methylation signature is specific for each cell-type, this approach enables to infer cell death in various settings. Based on this approach we developed a quantitative, cell specific circulating cfDNA markers for brain injury [25, 27]. In contrast to approaches designed to identify circulating tumor DNA [28], this approach does not rely on genetic differences between the host and the cancerous tissue of interest, and hence can detect death of genetically normal cells [29]. As radiotherapy is hypothesized to induce brain cells apoptosis as well as BBB disruption, cell free DNA fragments derived from neurons and glia, collectively termed brain-derived circulating free DNA (BncfDNA), are expected to be released to the circulation, and hence be identified by specific methylation signatures. In this report we describe BncfDNA dynamics throughout radiotherapy administered for the treatment of brain metastases, in the context of RBI.

## Methods

### Patient population

24 adult patients with one or more brain metastases scheduled for brain radiotherapy were studied between 31-1-2017 to 14-02-2019. Eligibility criteria included: (1) Age>18; (2) extra-CNS malignant tumor presenting with one or more brain metastases; (3) candidate for brain radiotherapy. Intended follow-up was up to 12 months after radiotherapy or until death or withdrawal, if sooner. All patients gave written informed consent for participation and the

study is approved by our institutional ethics committee, #0346–12 allowing identification of individual participants during or after data collection.

## Definition criteria of RBI and non-RBI tumor status

i.  Events of brain-radiotherapy related effects not related to systemic disease and with no evidence for tumor progression were defined as acute, early-delay and late-delayed RBI according to the clinical, radiological and temporal criteria [5, 6, 9, 18, 19, 30] as detailed in S3 Table.

ii.  Other events with no clear evidence of RBI were defined as non-RBI tumor status events and were sub-classified to progressive disease (PD), tumor response (TR) or stable disease (SD) according to clinical and radiological criteria detailed in S4 Table.

iii.  Cases of clinical deterioration associated with imaging findings equivocal for distinguishing between tumor progression, RBI, or combination of both were classified as undetermined radiological progression (URP).

Of note, definition of RBI was based solely on clinical and radiological criteria, as none of the patients underwent biopsy to confirm the diagnosis of RBI or tumor progression.

Clinical and imaging data including relevant time course for each patient are detailed in the supplementary appendix (S1 and S2 Tables and S1 Fig). Examples for characteristic imaging of RBI are provided in S9 Fig.

## Medical records and additional assessment

Treating physician and neuro-oncological visits, prescribed treatments, demographic data, and MRI studies were recorded on the Hadassah Medical Center electronic medical chart during the observation period. Longitudinal blood samples for DNA analysis were collected during and after radiotherapy until the end of follow-up (S5 Table). In this study we made an effort to be objective as possible in analysis and interpreting the data. For this purpose, blood samples were collected during patient visits, whenever feasible, irrespective of whether they presented with symptoms or not. Subsequently these blood samples were processed and analyzed independently, while maintaining blind to the clinical data. The majority of the medical records were routinely completed by physicians during admission and routine follow-up, which were not inherently related to the study, and collected later by research team unaffiliated with processing and analysis of blood samples. On the next step we coordinated between clinical and radiological characteristics of RBI to measured values of bncfDNA for further analysis.

## DNA samples

Blood samples were collected in EDTA tubes. CfDNA was extracted from 4 ml of plasma using the QI symphony System (Qiagen). Isolated cfDNA concentration was measured using Qubit dsDNA Kit (Invitrogen) and was treated with bisulfite (Zymo research). 20 nanograms of DNA from each sample were used for multiplex PCR amplification of 10 brain-specific methylation markers, followed by sequencing on an Illumina NextSeq machine.

## cfDNA methylation analysis

As previously reported[24, 25], bisulfite-treated cfDNA was PCR amplified using primers specific for bisulfite-treated DNA, but independent of methylation status at monitored CpG sites. Barcoded primers allowed for the sequencing of several samples from different individuals simultaneously on the NextSeq 500 (Illumina). The barcoded primers flanked the 6–9 CpG

sites and amplified a sequence shorter than 150 bp, allowing for amplification of cfDNA. Sequencing of the PCR products were conducted using NextSeq Reagent Kit v2 (Illumina method). Sequenced reads were separated by barcodes, aligned to the target sequence, and analyzed using custom scripts written and implemented in MATLAB. Reads were quality-filtered based on Illumina quality scores. Reads were identified by having at least 80% similarity to target sequences and containing all the expected CpGs in the sequence. CpGs were considered methylated if "CG" was read and considered unmethylated if "TG" was read. Efficiency of bisulfite conversion were assessed by analyzing methylation of non-CpG cytosines.

## Tissue-specific cfDNA versus absolute levels of cfDNA

To calculate the concentration of cfDNA derived from a specific tissue, the fraction of cfDNA derived from the tissue (as determined from the frequency of molecules carrying a tissue-specific methylation pattern) was multiplied by the concentration of cfDNA measured in the plasma of each patient.

## Brain specific methylation markers

10 brain-specific markers were utilized for this study. These were genomic loci that we have found to be unmethylated specifically in astrocytes (adjacent to the WOX, AST1 and PRDM2 genes), oligodendrocytes (adjacent to the NMR, TAF8, ZFP genes), or neurons (509, ITF, SLC, ZNF238). The development and validation of these markers were recently published [27]. Each marker was scored as the fraction of unmethylated molecules in a sample. Methylation markers of the same cell type were averaged to generate a cell type-specific score per sample. The summation of values for all brain tissue types was represented by total bncfDNA.

## Data analysis

Changes in clinical and imaging parameters defining RBI or non-RBI were correlated with the concurrent changes (an increase or decrease) in measured bncfDNA levels. For this purpose, each patient was analyzed individually, comparing measured bncfDNA levels in specific time points to his previous measured levels. Fold change was calculated as the ratio of bncfDNA levels measured during RBI or non-RBI event to its levels measured in the antecedent time point and represented in logarithmic scale (log2).

# Results

## Patient population

24 patients (98 samples) were enrolled for this study after signing an informed consent. Patient characteristics included: mean age of 60 years (25–79), 9 men and 15 women. Most common primary tumor were breast (6 patients), lung (5 patients), melanoma (3 patients), ovarian (2 patients), and colon (2 patients). Sixteen patients were treated with whole brain radiotherapy (WBRT) and 8 patients were treated with stereotactic radiosurgery (SRS).

## Clinical outcomes

During follow-up 9 RBI events occurred: 4 acute, 3 early-delayed, and 2 late-delayed. In 3 patients acute RBI resolved (S1 Table). Non-RBI events occurred in 15 patients: 3 progressive disease; 7 tumor response and 5 stable disease. In addition, in 4 patients an undetermined radiological progression (URP) occurred, (S2 Table). Patient status as RBI, non-RBI or URP was classified accordingly during each follow-up visit (S1 Fig).

## bncfDNA levels increase in RBI and decrease following acute RBI resolution

Nine patients suffered from RBI. Clinical presentation is correlated with a marked increase of total bncfDNA before and during RBI in 8 patients, and mild increase in 1 patient (Figs 1A and 1C and S2–S4 and S10). Interestingly, in 3 patients gradual elevation in bncfDNA precedes the clinical presentation of acute RBI (Figs 1A and S2; patients 1, 10, 11), suggesting that bncfDNA elevation precedes the onset of acute RBI before it is clinically evident. Clinical improvement of acute RBI symptoms, documented in 3 patients, is correlated with a decrease in total bncfDNA (Figs 1D and S2).

## BncfDNA levels are variable in non-RBI events

In 15 non-RBI events (3 progressive disease, 7 tumor response and 5 stable disease) changes of bncfDNA levels are independent of tumor status (Fig 1B and 1E). In the subgroup with tumor response increase or decrease in bncfDNA is dependent on time of measurement. In 3 patients demonstrating tumor response during radiotherapy course, there is a decrease in astrocyte-derived, neuron-derived, oligodendrocyte-derived, and total bncfDNA (Figs 1E and S5; patients 4, 12, 16). In contrast, in 4 patients demonstrating tumor response after completion of radiotherapy there is an increase in the levels of both astrocyte-derived, neuron-derived, and oligodendrocyte-derived and total bncfDNA (Figs 1E and S5; patients 5, 17, 18, 20). In events of stable disease, the trend of bncfDNA levels is variable between patients, independent of irradiation type or measurement time (Fig 1E and S6; patients 3,9,15,18,22). Three patients developed progressive disease during the follow up period. In all of them there was a decrease in total bncfDNA at progressive disease. (Figs 1E and S7; patients 6,22,23). All presentations of progressive disease occurred more than 9 months after radiotherapy, a timeframe in which early-delayed RBI is expected to resolve [5, 6, 30].

## Undetermined radiological progression is associated with increase in bncfDNA levels

Four patients presented with clinical and radiological progression of undetermined definition. In all 4 patients there is a positive correlation between radiological progression and an increase in total and all 3 cell-types bncfDNA, which might reflect either progressive disease or RBI, as well as the combination of both (Figs 1E and S8).

## Discussion

In this proof-of-concept study we aimed to determine if bncfDNA can be used to detect and monitor radiation related brain tissue injury among patients with brain metastases. We followed 24 patients before, during and after radiotherapy clinically, radiologically (MRI) and by sequential measurement of plasma derived bncfDNA. We found that bncfDNA elevation characterizes radiation-induced brain injury, both acute, early and late-delayed, as defined by clinical and radiological criteria. In 3 patients, increased levels of bncfDNA preceded the presentation of RBI (acute and early-delayed respectively), suggesting that bncfDNA elevation also reflects asymptomatic RBI and can indicate RBI before it is clinically evident. Thus, we suggest that bncfDNA may serve as a biomarker for early detection of RBI, both symptomatic and asymptomatic. In addition, decreased bncfDNA levels, reflecting cfDNA clearance, characterize resolution of RBI. We also found that changes in bncfDNA are not consistent in non-RBI events. In tumor response and stable disease bncfDNA both rise and drop, while progressive disease is associated with a decrease in bncfDNA levels. Interestingly, increase in

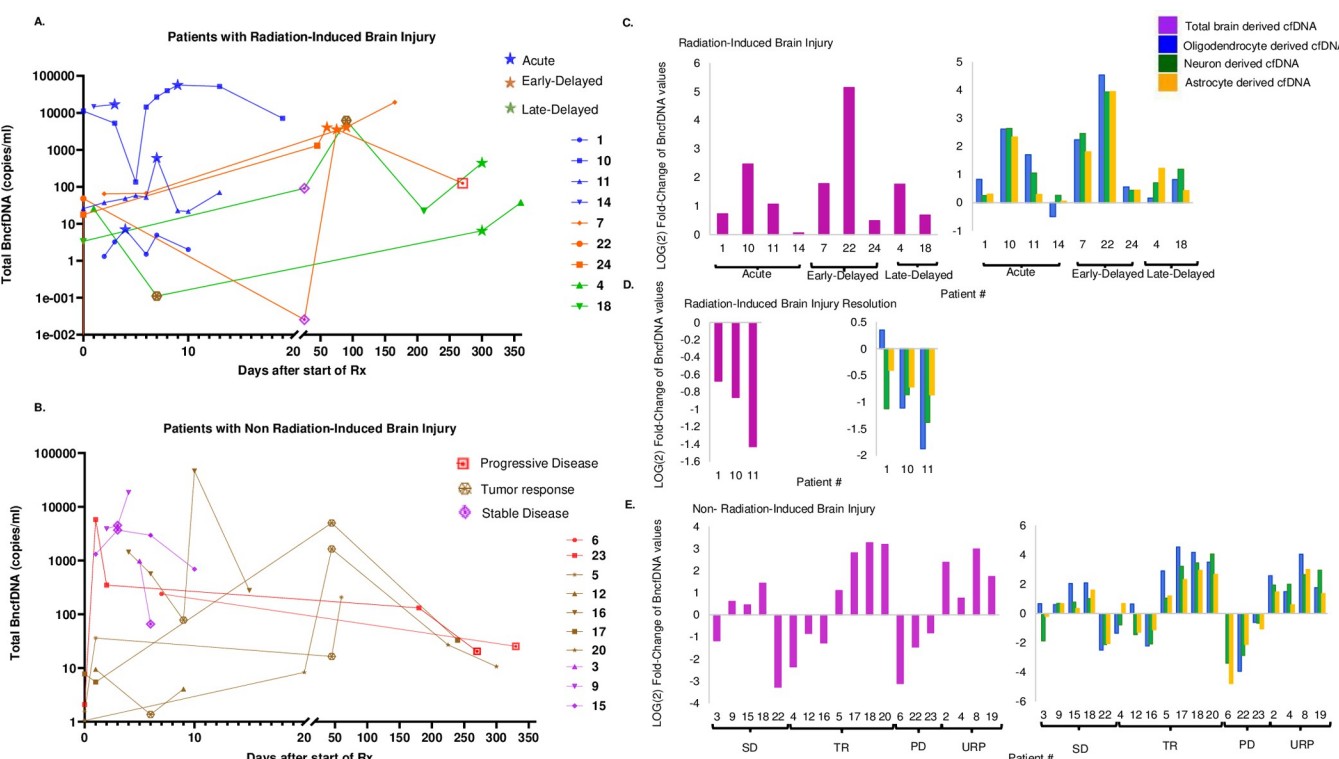

**Fig 1. BncfDNA levels dynamic during RBI and non-RBI.** Total BncfDNA levels measured for each patient (in log10 scale) during follow-up period for patients with RBI [A] and non-RBI [B]. Clinical onset of RBI is represented by a star (acute RBI–blue, early delayed–orange, late delayed–green). Clinical presentation of non-RBI is marked as detailed in the legend (progressive disease–red square, tumor response–brown hexagon; stable disease–purple rhomboid;). In patient 4, 18 with RBI, additional time-points of non-RBI clinical onset are marked in accordance to Fig 1B, as mentioned above. Fold change (log 2 scale) of bncfDNA levels in 9 patients during RBI onset [C], 3 patients during resolution of RBI [D], and 15 patients during non-RBI events [E]. Total bncfDNA marked in purple represent the mean summation of all cell-specific bncfDNA; oligodendrocytes (blue); Neuron (green) and astrocytes (Yellow). In the Y-Axis fold change as calculated: the ratio in bncfDNA levels measured during RBI, RBI resolution or non-RBI and the levels measured in the antecedent time point. BncfDNA: brain-derived circulating DNA SD: stable disease; PD: Progressive disease; TR: Tumor response; URP: Undetermined radiological progression.

bncfDNA levels is common to blood samples collected 45–90 days after brain radiotherapy, either if it is associated with RBI or tumor response. As this time period corresponds to the typical timeframe for early-delayed RBI appearance [5, 6, 30], we suggest that elevation of bncfDNA levels in those patients might reflect asymptomatic RBI [31]. The consistent increase of bncfDNA in RBI and decrease in progressive disease suggests its potential to differentiate between radiotherapy effects and tumor progression. However, additional mechanisms for decreasing levels of bncfDNA during progressive disease might exist.

Theoretically, enlargement of brain metastases may result in elevation of bncfDNA, as brain metastases may induce peritumoral tissue injury through mass effect, inflammatory cascade, or other possible mechanisms. In a separate work [27] we identified elevated neuron-, oligodendrocyte- and astrocyte-derived cfDNA in a small subpopulation of patients with pre-radiated brain metastases, while in most patients (26 of 29) brain metastases were not associated with significant elevated levels of bncfDNA compared with healthy individuals and cancer patients without brain metastases. In addition, no correlation is found between bncfDNA levels and other parameters in patients with pre-radiated brain metastases, including age, gender, recent neurosurgical intervention, and previous or recent chemotherapy exposure. While more work is needed to understand the relationship between brain metastases and brain-derived cfDNA, our findings suggest that bncfDNA levels reflect RBI better than they do brain

metastases progression. The level of total cfDNA reflects systemic tumor burden, that can increase or decrease independently of brain disease. Thus, compared to the ratio of bncfDNA to total cfDNA, absolute levels of bncfDNA may reflect RBI (S10 Fig). Brain-derived cfDNA levels depend both on the amount of brain cell injury, as well as its ability to enter the plasma. Thus, the pronounced increase in bncfDNA levels as measured in association with RBI in contrast to brain metastases progression may indicate a stronger BBB disruption induced by radiotoxicity [32, 33]. This also suggests the potential of bncfDNA to serve as a biomarker for BBB disruption to guide management in patients with brain metastases. However, additional mechanisms related to bncfDNA clearance, can also contribute to the individual variability of bncfDNA levels in pretreated patients as well as in non-RBI non-progressive tumor status. Thus, although decrease in bncfDNA levels during progressive disease might be related to a lack of brain cell injury in contrast to RBI, it is also might be attributed to differences in other mechanisms related to BBB disruption and/or cfDNA clearance from the plasma. Further studies are required to substantiate these ideas.

As differentiating tumor progression from pseudo-progression and radionecrosis is critically important for patients' management of primary and secondary brain tumors, several non-invasive strategies have been tested mainly in the context of the primary brain tumor glioblastoma multiforme (GBM), and are mainly based on tumor-derived circulating biomarkers to detect tumor progression [20–31, 33, 34]. However, all previously published methods, are based on the identification of tumor derived DNA, which in the case of brain metastases might be challenging owing to the broad molecular diversity among metastatic malignancies (variation by tumor histology and organ of origin with additional patient specific variation). In contrast to strategies using tumor DNA for detection of tumor progression, we suggest a method for detecting non-tumor DNA as a biomarker for normal tissue injury. As variations in brain-derived cfDNA levels appear to correlate with brain tissue damage, it can be used to detect treatment-related brain damage in the context of brain metastases. Moreover, the clear divergence in bncfDNA changes in RBI (increased levels) and progressive disease (decreased levels), suggests a potential utility of bncfDNA to help distinguish RBI from progressive disease in clinically ambiguous cases.

Given the very short half-life of cfDNA (estimated 15–120 minutes) [34], elevated levels of BncfDNA likely reflect active cell death of glia and neurons during RBI. Thus, in addition to its potential diagnostic utility, measurement of bncfDNA may suggest which brain cell types are being affected by brain radiotherapy at the time of sampling. Our results suggest the involvement of both neuronal cell apoptosis as well as glial cell death in all types of RBI, although in different proportions. Specifically, the most consistent elevation in acute RBI is related to neuron-derived bncfDNA, which might indicate a higher sensitivity to radiation damage of neuronal cells comparing to glial cells. In contrast, elevation of oligodendrocyte-derived cfDNA in different proportion between patients with acute RBI suggests for variable cellular radiosensitivity of oligodendrocytes between individualized patients. However, a more prominent and consistent increase in oligodendrocyte-derived cfDNA associated with early-delayed RBI and late-delayed RBI, supporting current knowledge attributing oligodendrocytes apoptosis an important role in the pathophysiology of these processes [9, 20, 31]. Moreover, as current hypotheses claim astrocyte proliferation and astrogliosis as a fundamental process induced by radiotherapy contributing to brain damage [9], our results indicating elevation of astrocyte-derived bncfDNA associated with RBI, suggest an induction of astrocyte cell death by acute and late radiation effects, as an additional component of RBI pathophysiological mechanism, requiring further investigation.

This pilot study is limited by the low number of patients tested, as well as by a lack of tissue biopsy to confirm diagnosis of tumor progression and RBI. Hence, larger studies will be

required to assess specificity and sensitivity of the various glial and neuronal bncfDNA, as well as the pathophysiological and clinical implications. However, there is a reason to believe that measurements of cfDNA methylation-based glial and neuronal cell deaths can be used as a diagnostic assay for early identification and monitoring of RBI, as well as differentiating radiation treatment effects from tumor progression.

## Conclusion

By measuring methylation markers of bncfDNA, we can identify neuron and glia cell death associated with RBI, both in symptomatic and asymptomatic patients. This quantitative minimally invasive method could be part of a novel biomarkers for early detection and monitoring RBI, with the potential to distinguish between tumor progression and pseudo-progression. Additional large-scale studies are required to further understand the biological background and clinical outcome of radiotherapy toxicity to the brain as well as to other organs.

## Supporting information

**S1 Fig. Individual patients'classification to RBI and non-RBI tumor status during follow-up.** Individual patients'classification to RBI and non-RBI tumor status during follow-up. During follow-up patient was classified to radiation-induced brain injury (RBI) or non-RBI tumor status according to clinical and radiological criteria. Each patient was able to be classified to more than one group in different time point along the follow-up period. Patients with no available clinical and radiographic data for analysis were excluded. RBI: radiotherapy induced brain injury; ED: early delayed; LD: late delayed; TR: tumor response; PD: progressive disease; SD: stable disease; URP: undetermined radiological progression; E: excluded.
(DOCX)

**S2 Fig. BncfDNA levels in ARBI and in ARBI resolution.** BncfDNA levels (copies/ml) during brain radiotherapy and follow up in 4 patients show significant increase correlating neurological worsening related to brain radiotoxicity (marked by red arrowhead), and bncfDNA decrease correlating clinical improvement (blue arrowhead). Each graph represents a different patient. Each colored line represents a specific tissue origin of bncfDNA as detailed in the key (astrocytes, neurons, oligodendrocytes). Total bncfDNA marked in purple represent the mean summation of all 3 tissue types' values. Mean baseline levels of bncfDNA among healthy individuals are: total bncfDNA (mean 1.32 copies/ml, std 3.2), astrocytes cfDNA (mean 1.76, std 5.4), oligodendrocytes cfDNA (mean 0.5, std 2.7), neurons cfDNA (mean 0.9, std 2.9). BncfDNA: brain-derived circulating DNA; RBI: radiation-induced brain Injury.
(DOCX)

**S3 Fig. BncfDNA levels in EDRBI.** BncfDNA levels (copies/ml) during the first 6 months after brain radiotherapy show significant increase in bncfDNA correlating to clinical and imaging manifestations of Early-Delayed RBI (marked by red arrowhead) in 3 Patients. Each graph represents a different patient. Each colored line represents a specific tissue origin of bncfDNA as detailed in the key (astrocytes, neurons, oligodendrocytes). Total bncfDNA marked in purple represent the mean summation of all 3 tissue types' values. Mean baseline levels of bncfDNA among healthy individuals are: total bncfDNA (mean 1.32 copies/ml, std 3.2), astrocytes cfDNA (mean 1.76, std 5.4), oligodendrocytes cfDNA (mean 0.5, std 2.7), neurons cfDNA (mean 0.9, std 2.9). BncfDNA: brain-derived circulating DNA; RBI: radiation-induced brain Injury.
(DOCX)

**S4 Fig. BncfDNA levels in LDRBI.** BncfDNA levels (copies/ml) during 300 days after brain radiotherapy show significant increase in bncfDNA correlating to clinical and imaging manifestations of Late Delayed RBI in 2 patients (marked by blue arrow). Each graph represents a different patient. Each colored line represents a specific tissue origin of bncfDNA as detailed in the key (astrocytes, neurons, oligodendrocytes). Total bncfDNA marked in purple represent the mean summation of all 3 tissue types' values. Mean baseline levels of bncfDNA among healthy individuals are: total bncfDNA (mean 1.32 copies/ml, std 3.2), astrocytes cfDNA (mean 1.76, std 5.4), oligodendrocytes cfDNA (mean 0.5, std 2.7), neurons cfDNA (mean 0.9, std 2.9). BncfDNA: brain-derived circulating DNA; RBI: radiation-induced brain injury.
(DOCX)

**S5 Fig. BncfDNA levels in TR.** BncfDNA levels (copies/ml) during follow-up after brain radiotherapy in 7 patients with tumor response (TR), marked by blue arrow. Each graph represents a different patient. Each colored line represents a specific tissue origin of bncfDNA as detailed in the key (astrocytes, neurons, oligodendrocytes). Total bncfDNA marked in purple represent the mean summation of all 3 tissue types' values. Mean baseline levels of bncfDNA among healthy individuals are: total bncfDNA (mean 1.32 copies/ml, std 3.2), astrocytes cfDNA (mean 1.76, std 5.4), oligodendrocytes cfDNA (mean 0.5, std 2.7), neurons cfDNA (mean 0.9, std 2.9). BncfDNA: brain-derived circulating DNA.
(DOCX)

**S6 Fig. BncfDNA levels in SD.** Documentation of stable clinical status in 5 patients is marked in a green arrowhead. Each colored line represents a specific tissue origin of bncfDNA as detailed in the key (astrocytes, neurons, oligodendrocytes). Total bncfDNA marked in purple represent the mean summation of all 3 tissue types of values. Mean baseline levels of bncfDNA among healthy individuals are: total bncfDNA (mean 1.32 copies/ml, std 3.2), astrocytes cfDNA (mean 1.76, std 5.4), oligodendrocytes cfDNA (mean 0.5, std 2.7), neurons cfDNA (mean 0.9, std 2.9). BncfDNA: brain-derived circulating DNA.
(DOCX)

**S7 Fig. BncfDNA levels in PD.** BncfDNA levels in progressive disease (PD) following brain radiotherapy. Documentation of PD in 3 patients is marked in a gray arrowhead. Each colored line represents a specific tissue origin of bncfDNA as detailed in the key (astrocytes, neurons, oligodendrocytes). Total bncfDNA marked in purple represent the mean summation of all 3 tissue types of values. Mean baseline levels of bncfDNA among healthy individuals are: total bncfDNA (mean 1.32 copies/ml, std 3.2), astrocytes cfDNA (mean 1.76, std 5.4), oligodendrocytes cfDNA (mean 0.5, std 2.7), neurons cfDNA (mean 0.9, std 2.9). BncfDNA: brain-derived circulating DNA.
(DOCX)

**S8 Fig. BncfDNA levels in URP.** BncfDNA levels in undetermined radiological progression (URP) following brain radiotherapy. Documentation of URP in 4 patients is marked in a brown arrowhead. Each colored line represents a specific tissue origin of bncfDNA as detailed in the key (astrocytes, neurons, oligodendrocytes). Total bncfDNA marked in purple represent the mean summation of all 3 tissue type values. Mean baseline levels of bncfDNA among healthy individuals are: total bncfDNA (mean 1.32 copies/ml, std 3.2), astrocytes cfDNA (mean 1.76, std 5.4), oligodendrocytes cfDNA (mean 0.5, std 2.7), neurons cfDNA (mean 0.9, std 2.9). BncfDNA: brain-derived circulating DNA.
(DOCX)

**S9 Fig. Imaging studies reflecting radiotherapy effects.** Imaging studies reflecting radiotherapy effects. MRI of patient 7 before and 3 months after WBRT (**A**) showing decrease of enhancing lesions on T1GAD (bottom, arrowhead), accompanied with increase in periventricular white matter changes on FLAIR (top, white arrow), reflecting early delayed RBI. MRI of patient 22 before SRS and 2 and 4 months later (**B**) showing progressive increase in enhancing lesion on TaGAD (top., white arrow), compatible with radiation effects reflected by red signal on TRAM (bottom, black arrow), thus reflecting early-delayed RBI. WBRT: whole brain radiotherapy; SRS: stereotactic radiosurgery.
(DOCX)

**S10 Fig. Relative total bncfDNA levels in RBI and non-RBI.** Ratio of total BncfDNA to total cfDNA levels measured for each patient during follow-up period, grouped to patients with RBI (blue lines) and non-RBI (red lines). Specific time points in which RBI and non-RBI occurred are marked as detailed in the legend (RIBD- blue star; treatment response–red hexagon; stable disease–red rhomboid; progressive disease–red square).
(DOCX)

**S1 Table. Individual clinical details of RBI cases.** Each case of radiation-induced brain injury (RBI) is detailed by the patient in which it occurred (patient #), the day after radiotherapy initiation in which it presented (presenting day), the radiotherapy modality, the clinical manifestation and imaging presentation correlated to the time of RBI occurrence. *Abbreviation*: WBRT, whole brain radiotherapy; SRS, stereotactic radiosurgery; TRAM, treatment response assessment map; NA, non-available.
(DOCX)

**S2 Table. Individual clinical details of cases related to non-RBI tumor status.** Each case of tumor status not related to RBI (response, progressive, stable) is detailed by the patient in which it occurred (patient #), the day after radiotherapy initiation in which it presented (presenting day), the radiotherapy modality, the clinical manifestation and imaging presentation correlated to the time of RBI occurrence. Cases of radiological progression which could not be solely attributed to RBI to tumor progression (URP) are also detailed as above. *Abbreviation*: RBI, radiation induced brain injury; PD, progressive disease; TR, tumor response; SD, stable disease; URP, undetermined radiological progression; LM, leptomeningeal; WBRT, whole brain radiotherapy; SRS, stereotactic radiosurgery; CSI, craniospinal irradiation; NA, non-available.
(DOCX)

**S3 Table. Criteria for RBI definition and classification.** Clinical, Radiological, and temporal criteria for definition and classification of radiotherapy-induced brain injury. *Abbreviation*: TRAM, treatment response assessment map.
(DOCX)

**S4 Table. Criteria for non RBI definition and classification.** Clinical and radiological criteria for definition and classification of non radiotherapy-induced brain injury (non RBI). *Abbreviation*: TRAM, treatment response assessment map.
(DOCX)

**S5 Table. Comprehensive patients data.**
(XLSX)

## Acknowledgments

The authors are grateful to the patients who participated in this study. We thank Ziva Inbar and Aviad Klinger for their great contribution in statistical analysis and graphs production; Dana Sheril Roffe and Aliza Moshayev for assisting with serum collection.

## Author Contributions

**Conceptualization:** Chen Makranz, Asael Lubotzky, Benjamin Glaser, Jonathan Cohen, Eli Sapir, Yuval Dor, Aviad Zick.

**Data curation:** Chen Makranz, Asael Lubotzky, Hai Zemmour, Ruth Shemer, Myriam Maoz, Eli Sapir, Aviad Zick.

**Formal analysis:** Chen Makranz, Aviad Zick.

**Funding acquisition:** Chen Makranz, Asael Lubotzky, Jonathan Cohen, Yuval Dor, Aviad Zick.

**Investigation:** Chen Makranz, Hai Zemmour, Ruth Shemer, Aviad Zick.

**Methodology:** Chen Makranz, Ruth Shemer, Alexander Lossos, Yuval Dor, Aviad Zick.

**Project administration:** Aviad Zick.

**Resources:** Marc Wygoda, Tamar Peretz, Jon Feldman.

**Supervision:** Tamar Peretz, Yuval Dor.

**Writing – original draft:** Chen Makranz, Aviad Zick.

**Writing – review & editing:** Chen Makranz, Asael Lubotzky, Jonathan Cohen, Eli Sapir, Noam Weizman, Ross A. Abrams, Alexander Lossos, Yuval Dor, Aviad Zick.

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
