## [Decision Letter · Decision Letter 0]

31 May 2023

PONE-D-23-10042Plasma based biomarkers detect radiation induced neurotoxicity in cancer patients treated for brain metastasis: A Pilot study.: A Pilot study.

Dear Dr. Zick,

Thank you for submitting your manuscript to PLOS ONE. After careful consideration, we feel that it has merit but does not fully meet PLOS ONE’s publication criteria as it currently stands. Therefore, we invite you to submit a revised version of the manuscript that addresses the points raised during the review process. Please revise this manuscript carefully according to the reviewer's suggestion.

We look forward to receiving your revised manuscript.

Kind regards,

Ryuya Yamanaka, M.D.,Ph.D.

Academic Editor

PLOS ONE

2. Please amend either the title on the online submission form (via Edit Submission) or the title in the manuscript so that they are identical.

Reviewers' comments:

Reviewer's Responses to Questions

**Comments to the Author**

1. Is the manuscript technically sound, and do the data support the conclusions?

Reviewer #1: Partly

Reviewer #2: Yes

2. Has the statistical analysis been performed appropriately and rigorously? 

Reviewer #1: No

Reviewer #2: Yes

3. Have the authors made all data underlying the findings in their manuscript fully available?

Reviewer #1: No

Reviewer #2: Yes

4. Is the manuscript presented in an intelligible fashion and written in standard English?

Reviewer #1: Yes

Reviewer #2: Yes

5. Review Comments to the Author

Reviewer #1: SUMMARY

In this study the authors use circulating DNA analyzed with respect to specific methylation patterns to deconvolute the amount of normal brain tissue damage during radiation therapy. The use of methylation-specific patterns is an extremely exciting, potentially powerful, blood-based tool and biomarker to differentiate the origin of different cells within the brain and associated damage. Furthermore, our understanding of neurotoxicity and radiation is poorly understood. Therefore, this study brings a very exciting technology to better understand radiation-induced neurotoxicity. Although a very exciting and timely premise, there are several key framing and data flaws that must be addressed before it becomes clear whether the findings and conclusions of the study are supported by the data. As currently written, the paper appears more as a set of case reports with limited statistical validity and may be better suited as a short report. The data does not appear objectively presented and there is no clear objective comparison of “controls” and “events”. The concern is that notable RIN events were cherry picked, and then only the compelling data are shown. Although the overall premise is very exciting and intriguing, the presented data does not allow for objective comparisons to confirm that this phenomenon is real. For example, more “negative” cases or data from patients during an absence of symptoms are needed to ensure that brain cfDNA is not fluctuating in general in these patients. Several of the patients have only 3 data points, which also limits understanding of the trajectory of brain cfDNA in these patients.

COMMENTS

A major issue with the paper is the terminology, which is imprecise and lumps many likely distinct phenomena together under the term Radiation Induced Neurotoxicity (RIN). For example, pseudoprogression is cited as an early form of RIN. I think most clinicians would agree that pseudoprogression refers to enlargement/radiographic worsening of existing disease sites and may indicate ongoing treatment response, which may reflect not true toxicity (neuro cell death) but might be more in line with an exaggerated tumor response and associated inflammation. In addition, this study relies on a concept of acute neurotoxicity based on “two” references (REF 5 and 6). However, REF5 is from 2000, and REF6 just references REF5 with no further elaboration. Therefore this is a rather dated concept. Given the above, it would be far more useful for the authors to adopt a more agnostic approach to “RIN”. The authors need to clearly define how they are using this term and what it encompasses. For example – do they mean symptomatic decrease in cognitive function or neurological deficits? Does it include just radiologic deficits (pseudoprogression can be either radiologic only (no symptoms), clinical only, or both). I suspect the authors are implying symptomatic neurocognitive deficits – but the term/concept needs to be precisely defined. This is not just a semantic issue – it is critical to how they conducted/collected and analyzed the data? My concern is that in the absence of a clear definition of RIN that was prospectively applied, the potential for cherry picking cases and identifying spurious associations is high. This concern is supported by table S6 - which shows that for the acute RIN cases, the symptoms are quite profound and occurred within 3-9 days of WBRT, and likely reflect inflammation/treatment response/pseudoprogression within the treated tumor as opposed to neurotoxicity of normal tissue caused directly by the radiation. These points are addressable but require careful consideration. The authors may benefit from the perspective of a neuro radiation oncologist if they have not already to help ensure an appropriate clinical perspective.

The actual data presented appear quite limited. The figure shows data from 5 cases, again which is concerning for cherry picking cases that align with the investigators hypotheses. Examination of individual cases might reflect any number of phenomena. For example, progression of disease, stroke, or chemo could all impact the brain damage and might have influenced the observations of patient 10. In order to remedy this, the data for all patients should be ideally shown alongside one another anchored at the same time point and would ideally show both absolute and relative levels of BnctDNA, since many patients may have elevated levels at baseline due to damage from tumor, chemo, infection/sepsis, or other phenomena.

It would be best to show all patients – both those with and without neurotoxicity, alongside one another so that a more objective display of the data can be appreciated.

It seems that all the subtypes of brain derived cfDNA mirror one another. Although it is important to show this in the supplementary data, it may be better to focus on just the total amount so that more patients can be represented within the same plot/graph so that we can see a more objective comparison between those that did vs. did not have RIN.

In table S1 – the details of individual RIN cases – the clinical manifestations are quite striking. For example all the acute RIN (ARIN) cases have symptoms such as cerebellar signs and somnolence, weakness and inability to talk, central facial palsy, and dizziness and visual disturbances – all within 3-9 days of starting WBRT. These seem quite extreme to be expected toxicities of WBRT – I would assume that all of these patients had profound disease burden and that this is related to acute tumor swelling vs. direct toxicity of the radiation. It may be helpful to hear more about the clinical details of these patients, since these side effects are not likely due to the XRT itself.

In the supplemental data (for example S6) the legend says that total BnctDNA is composed of four cell types, but only three cell types are shown. Also – it appears the mean “total” brain cDNA is less than some of the individual components, which doesn’t seem like it should be possible if total is a summation of each individual type.

It would be critical to know what other clinical events happened for patients – particularly initiation of systemic therapy.

Reviewer #2: First, I want to congratulate and acknowledge the work done by the entire team on this project.

Radionecrosis, and the more encompassing radiation-induced neurotoxicity, is a very challenging clinical scenario that clinicians encounter daily. The identification of a blood-based biomarker that can, with adequate sensitivity and specificity, identify radiation-induced neurotoxicity would be practice changing.

The methodology of the testing appears sound, but as a clinician, I will have to defer to a reviewer with more nuanced understanding of these tests.

In respect to the clinical aspect, my main concern is the defining of Acute RIN, early-delayed RIN, and late-delayed RIN. While the temporal definitions are explicit in the text, there is no explanation of what clinical or radiographic findings signified a neurotoxic event. Especially with neuro symptoms which are inherently more subjective, it is important to discern the toxicity.

I was surprised that >33% patients experienced RIN. That seems to be high compared to historical comparisons. Furthermore, four of the 16 patients (25%) had acute RIN with WBRT. Rarely is imaging performed routinely in this time, so would that be clinical deterioration? And how is that determined vs disease progression.

My other questions, were the blood draws for the cfDNA standardized? Why were both WBRT and SRS considered as there are two vastly different patient populations (from performance status) and treatment approaches.

As described above, kudos to the team on this work. It is an important proof-of-concept I hope can be further built upon.

6. PLOS authors have the option to publish the peer review history of their article (what does this mean?). If published, this will include your full peer review and any attached files.

Reviewer #1: No

Reviewer #2: No

---

## [Author Response · Author response to Decision Letter 0]

17 Jul 2023

Response to Reviewers:

Dear Reviewers,

Thank you for reviewing our manuscript. We appreciate your comments and helpful suggestions. We revised our paper accordingly. First, we submit our work as short report as suggested. In response to the concerns mentioned we revised our terminology to the updated term “radiotherapy induced brain disease” that refers to radiation induced toxicity effecting not only neurons but various types of brain cells including glia (instead of radiation induced neurotoxicity) and added the more recently published paper on this issue. We also detailed the criteria used to define cases of radiotherapy induced complication and other presentations not related to radiotherapy effects. Additionally, we added a table summarizing clinical information and bncfDNA levels of all patients in this study. 

These changes and others are summarized below (marked in red color below each comment). Pages and line numbers (marked in blue color) refer to the version of “revised manuscript highlighted with track changes”.

1. Is the manuscript technically sound, and do the data support the conclusions?

Reviewer #1: Partly

We corrected the manuscript according to the suggestion in the reviewer summary (detailed below)

Reviewer #2: Yes

2. Has the statistical analysis been performed appropriately and rigorously?

Reviewer #1: No

The small number of participants in this pilot study does not enable an extensive statistical analysis, hence results can be interpreted qualitatively. As suggested below we revised the manuscript as a short report accordingly.

Reviewer #2:

 Yes

3. Have the authors made all data underlying the findings in their manuscript fully available?

Reviewer #1: No

In order to make all data available we added to the supporting information appendix in the revised submission with detailed clinical and radiological data collected for each patient, coordinated with the measured levels of bncfDNA from timely blood samples (Table S5). 

Reviewer #2: Yes

4. Is the manuscript presented in an intelligible fashion and written in standard English?

Reviewer #1: Yes

Reviewer #2: Yes

5. Review Comments to the Author

Reviewer #1: SUMMARY

In this study the authors use circulating DNA analyzed with respect to specific methylation patterns to deconvolute the amount of normal brain tissue damage during radiation therapy. The use of methylation-specific patterns is an extremely exciting, potentially powerful, blood-based tool and biomarker to differentiate the origin of different cells within the brain and associated damage. Furthermore, our understanding of neurotoxicity and radiation is poorly understood. Therefore, this study brings a very exciting technology to better understand radiation-induced neurotoxicity. Although a very exciting and timely premise, there are several key framing and data flaws that must be addressed before it becomes clear whether the findings and conclusions of the study are supported by the data. As currently written, the paper appears more as a set of case reports with limited statistical validity and may be better suited as a short report. 

As suggested, we revised the manuscript as a short report. For that purpose, we made some changes in the text to make is shorter and focus on the main findings, as below:

1. Page 4-5, lines 76-94: We deleted the definition criteria for RIBD and non-RIBD sub-classes, and referred the reader to 2 tables in the supporting information appendix that describe now in more details the definition criteria for each subclass (table 3 for RIBD subclassification and table 4 for non-RIBD subclassification)

2. Page 7, line 140-141: We deleted the paragraph specifying the number of patients treated with SRS compared to WBRT, as it is not critical information for results interpretation.

3. Page 9, line 190: We deleted the paragraph specifying number of patients treated with SRS compared to WBRT, as it is not a critical information for results interpretation.

4. Page 9, line 191-192 and line 196-199: We omitted the quantitative information of specific bncfDNA changes and referred the reader to the relevant figures in the supporting information appendix which represents this quantitative information.

5. Page 10, line 218-220: We omitted the quantitative information of specific bncfDNA changes and referred the reader to the relevant figures in the supporting information appendix which represents this quantitative information.

6. Page 11, line 228-230: We omitted the quantitative information of specific bncfDNA changes and referred the reader to the relevant figures in the supporting information appendix which represents these quantitative information.

7. Page 11, line 226-227: We omitted a paragraph referring to patient 15 as it does not represent the whole group and no firm conclusions can be made.

8. Page 11, line 232-243 : We deleted legend of figure 2, since we omitted figure 2 and combined data of both RIBD and non-RIBD in single figure (fig 1) as suggested by reviewer 1 (see also below). 

9. Page 13, line 243-251: We omitted the quantitative information of specific bncfDNA changes and referred the reader to the relevant figures in the supporting information appendix which represents this quantitative information.

The data does not appear objectively presented and there is no clear objective comparison of “controls” and “events”. The concern is that notable RIN events were cherry picked, and then only the compelling data are shown. Although the overall premise is very exciting and intriguing, the presented data does not allow for objective comparisons to confirm that this phenomenon is real. For example, more “negative” cases or data from patients during an absence of symptoms are needed to ensure that brain cfDNA is not fluctuating in general in these patients. Several of the patients have only 3 data points, which also limits understanding of the trajectory of brain cfDNA in these patients.

In this study we made an effort to be objective as possible in analysis and interpreting the data. However, from your comments we understand that we might not explain our approach clearly enough. Hence, we added in the method section a paragraph referring to this point (page 5, line 102-108). For the purpose of objective data analysis and interpretation - blood samples were collected during patient visits (whenever feasible) regardless of if they had symptoms or not, and those blood samples were processed and analyzed independently and blinded to the clinical data. Most of the medical records were completed as part of the routine/admission follow-up by physicians which were not necessarily related to study and collected later by research team unrelated to blood samples analysis. On the next step we coordinated between clinical and radiological characteristics of RIBD to measured values of bncfDNA for further analysis.. This approach excluded the possibility of biased collection of patient data by clinical staff. However, it also resulted in missing clinical data in several time-points in which of blood samples were collected, especially when no symptomatic neurological deterioration occurred. For that reason, for some patients we have very few time points which could referred to stable/no symptoms in which we also have compatible blood samples. These cases were referred to a specific group of cases named “stable disease (SD) and shown in table S2 and figure S6, and discussed in the results section (page 11, line 225-226). Due to the low number of cases in this group we cannot make a statistical meaningful comparison between “events” and “controls” in the current pilot study.

COMMENTS

A major issue with the paper is the terminology, which is imprecise and lumps many likely distinct phenomena together under the term Radiation Induced Neurotoxicity (RIN). For example, pseudoprogression is cited as an early form of RIN. I think most clinicians would agree that pseudoprogression refers to enlargement/radiographic worsening of existing disease sites and may indicate ongoing treatment response, which may reflect not true toxicity (neuro cell death) but might be more in line with an exaggerated tumor response and associated inflammation. In addition, this study relies on a concept of acute neurotoxicity based on “two” references (REF 5 and 6). However, REF5 is from 2000, and REF6 just references REF5 with no further elaboration. Therefore this is a rather dated concept. Given the above, it would be far more useful for the authors to adopt a more agnostic approach to “RIN”. 

We understand that the terminology we used was confusing and not accurate. To correct this point - In the revised paper, we replaced the term “radiation induced radiotoxicity” with the term “radiation induced brain disease”, as described in recent review by Gorbunov and Kiang[1], to encompass both direct toxicity to neuron and glia cells by radiation , as well as indirect effects of radiation on neuronal and glia cell mediated by vascular damage and neuroinflammation. This paper was added to the manuscript as reference 7 (page 3, line 25-28). All “RIN” terms replaced with “RIBD”, and all “non-RIN” replaced with “non-RIBD” throughout the manuscript, as well as in the tables and figures (including in the supporting information appendix).

The authors need to clearly define how they are using this term and what it encompasses. For example – do they mean symptomatic decrease in cognitive function or neurological deficits? Does it include just radiologic deficits (pseudoprogression can be either radiologic only (no symptoms), clinical only, or both). I suspect the authors are implying symptomatic neurocognitive deficits – but the term/concept needs to be precisely defined. This is not just a semantic issue – it is critical to how they conducted/collected and analyzed the data? My concern is that in the absence of a clear definition of RIN that was prospectively applied, the potential for cherry picking cases and identifying spurious associations is high.

Following reviewers comments we added to the supporting information appendix of the revised submission table S3, which details the specific clinical and radiological criteria used to define acute, early delayed or late radiotherapy related complications, and table S4 that detailed our criteria definition of non-RIBD presentations of tumor progression, tumor response or stable disease. As shown in table S3 and table S2, patients with clinical or imaging presentation questionable to be radiotherapy induced and patients with a mix response of both radiation induced complication and tumor progression (for example patient 4 in day 365) were not classified as having radiotherapy induced complication, but classified to different group of “undetermined radiological progression” (URP). Hence, only cases that clearly complied definition criteria were included as radiotherapy induced brain disease.

This concern is supported by table S6 - which shows that for the acute RIN cases, the symptoms are quite profound and occurred within 3-9 days of WBRT, and likely reflect inflammation/treatment response/pseudoprogression within the treated tumor as opposed to neurotoxicity of normal tissue caused directly by the radiation. These points are addressable but require careful consideration. 

We agree that most acute RIBD occurred in first 10 days of treatment and was quite profound, and is mostly a result of inflammatory response. To address this we would like to emphasize that most WBRT treated patients were admitted during radiotherapy due to the large burden of disease, which explains the high occurrence of striking neurological symptoms during WBRT. Hence, although our cohort of WBRT patients is biased for progressive disease, cases of radiotherapy induced brain disease were defined according to the strict criteria mentioned in table S3. Moreover, as mentioned above, although inflammatory response to radiotherapy is a part of the tumor response, it also effects indirectly neurons and glia, resulting in variety of neurological presentations, and hence also included here as radiotherapy-induced brain disease.

The authors may benefit from the perspective of a neuro radiation oncologist if they have not already to help ensure an appropriate clinical perspective.

All images were interpreted by a neuroradiologist as a part of the standard routine follow-up for all patients in our clinical center. Clinical perspective was also supported by 2 neuro-radiation oncologist which are co-authors in this paper (Dr. Eli Sapir and Dr. Mark Wigoda).

The actual data presented appear quite limited. The figure shows data from 5 cases, again which is concerning for cherry picking cases that align with the investigators hypotheses. Examination of individual cases might reflect any number of phenomena. For example, progression of disease, stroke, or chemo could all impact the brain damage and might have influenced the observations of patient 10. In order to remedy this, the data for all patients should be ideally shown alongside one another anchored at the same time point and would ideally show both absolute and relative levels of BnctDNA, since many patients may have elevated levels at baseline due to damage from tumor, chemo, infection/sepsis, or other phenomena.

It seems that all the subtypes of brain derived cfDNA mirror one another. Although it is important to show this in the supporting information data, it may be better to focus on just the total amount so that more patients can be represented within the same plot/graph so that we can see a more objective comparison between those that did vs. did not have RIN.

It would be best to show all patients – both those with and without neurotoxicity, alongside one another so that a more objective display of the data can be appreciated.

In response to this comment we change the main figure (Fig 1) and created Figure 1A that summarized absolute levels of total bncfDNA in all time points measured for each patient, as suggested by reviewer 1. However, it should be noted that some patients had both radiotherapy induced, and non-radiotherapy induced presentations in different time point during follow-up (for example patient 18 presented with radiotherapy induced brain disease on day 240, tumor response on day 90 and stable disease on day 26). Hence, We suggest to compare bncfDNA levels between time points relevant to each entity in each individual patient, and not necessarily between patients or groups. Additional new graph showing relative bncfDNA for all patients during follow up is given in Figure S10 in the supporting information appendix.

In table S1 – the details of individual RIN cases – the clinical manifestations are quite striking. For example all the acute RIN (ARIN) cases have symptoms such as cerebellar signs and somnolence, weakness and inability to talk, central facial palsy, and dizziness and visual disturbances – all within 3-9 days of starting WBRT. These seem quite extreme to be expected toxicities of WBRT – I would assume that all of these patients had profound disease burden and that this is related to acute tumor swelling vs. direct toxicity of the radiation. It may be helpful to hear more about the clinical details of these patients, since these side effects are not likely due to the XRT itself.

We agree with this comment. Indeed, most WBRT treated patients were admitted during radiotherapy due to the large burden of disease, which probably resulted in the extreme neurological symptoms presented as suggested by reviewer 1. However, although inflammatory response to radiotherapy is a part of the tumor response, it also effects indirectly neurons and glia, resulting in variety of neurological presentations, and hence also included here as radiotherapy-induced brain disease. More clinical data of each patient is supplied in the new excel sheet added as table S5.

In the supplemental data (for example S6) the legend says that total BnctDNA is composed of four cell types, but only three cell types are shown. Also – it appears the mean “total” brain cDNA is less than some of the individual components, which doesn’t seem like it should be possible if total is a summation of each individual type.

Thank you for this comment. This was a typo mistake and we corrected it in the new version to “3 cell types” in the figure legend of the relevant supplementary graphs S2-S8. We also recalculated total values of bncfDNA and corrected all relevant graphs accordingly.

It would be critical to know what other clinical events happened for patients – particularly initiation of systemic therapy.

We supplied this information in the new excel sheet added as table S5.

Reviewer #2: First, I want to congratulate and acknowledge the work done by the entire team on this project.

Radionecrosis, and the more encompassing radiation-induced neurotoxicity, is a very challenging clinical scenario that clinicians encounter daily. The identification of a blood-based biomarker that can, with adequate sensitivity and specificity, identify radiation-induced neurotoxicity would be practice changing.

The methodology of the testing appears sound, but as a clinician, I will have to defer to a reviewer with more nuanced understanding of these tests.

In respect to the clinical aspect, my main concern is the defining of Acute RIN, early-delayed RIN, and late-delayed RIN. While the temporal definitions are explicit in the text, there is no explanation of what clinical or radiographic findings signified a neurotoxic event. Especially with neuro symptoms which are inherently more subjective, it is important to discern the toxicity.

Following reviewers comments we added to the revised paper a table (table S3) to the supporting information appendix, which details the specific neurological presentations, as well as radiological criteria used to define acute, early delayed, or late radiotherapy related complications.

I was surprised that >33% patients experienced RIN. That seems to be high compared to historical comparisons. Furthermore, four of the 16 patients (25%) had acute RIN with WBRT. Rarely is imaging performed routinely in this time, so would that be clinical deterioration? And how is that determined vs disease progression.

Most WBRT treated patients were admitted during radiotherapy due to the large burden of disease, which may explain the high occurrence of striking neurological symptoms during WBRT. Determining those presentations as radiotherapy related vs disease progression was based on the criteria detailed in table S3. 

My other questions, were the blood draws for the cfDNA standardized?

Values of bncfDNA were standardized according to total cfDNA measured in blood. In previous study we published[2] we have shown absolute levels of BncfDNA in healthy individuals and in cancer patients with and without brain metastasis. However, assessment and analysis of the BncfDNA levels in this study were not compared by their absolute levels between groups, but for each individual patient we compared the relative changes in BncfDNA levels between different time-points.

 Why were both WBRT and SRS considered as there are two vastly different patient populations (from performance status) and treatment approaches.

We differentiated between SRS and WBRT treated patients since each of the radiation treatment is different in the mechanism and risk for radiation induced brain complications, due to difference in fraction dose, number of fractions, total dose and radiation field. However, in view of this comment, and since in our cohort no clear difference between those 2 treatments can be clearly concluded, we omitted this differentiation between patients in the revised manuscript. (Page 7, line 140-141, Page 9, line 190).

As described above, kudos to the team on this work. It is an important proof-of-concept I hope can be further built upon.

6. PLOS authors have the option to publish the peer review history of their article (what does this mean?). If published, this will include your full peer review and any attached files.

Do you want your identity to be public for this peer review? For information about this choice, including consent withdrawal, please see our Privacy Policy.

Reviewer #1: No

Reviewer #2: No

References:

1. Gorbunov, N.V. and J.G. Kiang, Brain Damage and Patterns of Neurovascular Disorder after Ionizing Irradiation. Complications in Radiotherapy and Radiation Combined Injury. Radiat Res, 2021. 196(1): p. 1-16.

2. Lubotzky, A., et al., Liquid biopsy reveals collateral tissue damage in cancer. JCI Insight, 2022. 7(2).

---

## [Decision Letter · Decision Letter 1]

11 Aug 2023

PONE-D-23-10042R1Short Report: Plasma based biomarkers detect radiation induced neurotoxicity in cancer patients treated for brain metastasis: A Pilot study.PLOS ONE

Dear Dr. Zick,

Thank you for submitting your manuscript to PLOS ONE. After careful consideration, we feel that it has merit but does not fully meet PLOS ONE’s publication criteria as it currently stands. Therefore, we invite you to submit a revised version of the manuscript that addresses the points raised during the review process.

We look forward to receiving your revised manuscript.

Kind regards,

Ryuya Yamanaka, M.D.,Ph.D.

Academic Editor

PLOS ONE

Reviewers' comments:

Reviewer's Responses to Questions

**Comments to the Author**

1. If the authors have adequately addressed your comments raised in a previous round of review and you feel that this manuscript is now acceptable for publication, you may indicate that here to bypass the “Comments to the Author” section, enter your conflict of interest statement in the “Confidential to Editor” section, and submit your "Accept" recommendation.

Reviewer #1: (No Response)

Reviewer #3: (No Response)

2. Is the manuscript technically sound, and do the data support the conclusions?

Reviewer #1: Partly

Reviewer #3: Yes

3. Has the statistical analysis been performed appropriately and rigorously? 

Reviewer #1: N/A

Reviewer #3: Yes

4. Have the authors made all data underlying the findings in their manuscript fully available?

Reviewer #1: Yes

Reviewer #3: Yes

5. Is the manuscript presented in an intelligible fashion and written in standard English?

Reviewer #1: Yes

Reviewer #3: No

6. Review Comments to the Author

Reviewer #1: The authors have responded thoroughly to the reviews. The work remains intriguing and novel. There are a few issues that remain that need to be addressed prior to acceptance for publication.

Figure 1A looks quite busy and the pattern / association with RIN/RIBD is not clear and does not support the idea that increases in cfDNA tightly correlate with brain injury. However, this may be due to noise and variation inherent to the data. Perhaps two figures would be better. One panel could look only at XRT patients without RIBD (red lines), anchored by start of XRT (align all the curves with respect to clinical onset of RIBD instead of XRT start). The RIBD panel (blue lines) might be better anchored by the date of RIBD. The patterns are just quite difficult to see in the figure 1A, which looks quite noisy, and look like they argue against any real pattern, but I suspect this can be teased out using an approach something like the above.

Adding the detailed individual case courses is very helpful. Thank you for reporting these data.

For patient #11 WBRT – they appeared to have a single spike in cfDNA – did their symptoms only last a few days and then resolve? Or did they persist after cfDNA levels returned to normal? This is worth a brief comment/discussion.

The revisiting of the terminology is appreciated. However, in the cited paper the term “Radiation-Induced Brain Injury” appears to be used far more often than “Radiation-Induced Brain Disease” (17 vs. 3 times). The term “disease” implies a more sustained, potentially permanent condition and physiologic entity, whereas the term “Radiation Inducted Brain Injury” would be perfect and corresponds nicely to what the authors are directly claiming to measure – injury of brain cells (both neurons and glia). The term “Radiation-Induced Brain Injury” should be used throughout to avoid confusion or a pejorative connotation.

Minor comments:

Table 1 is quite simplistic and these data are probably better just described in the text, separating the descriptions of the SRS vs. WBRT patients with respect to timing of RIBD events (these are very different clinical contexts).

Table S1: The definition column might be more clear to just state “Acute, Early-Delayed, or Late Delayed” instead of acronyms like ARIBD, EDRIBD, LDRBID.

Table S2: Can an alternative to “URP” like “PD” be use for progressive disease?

Table S4: Does the X axis indicate time? Is this to scale? If so time should be added. The figure is helpful for spelling out the clinical course of each patient.

Reviewer #3: The authors are to be applauded for seeking to identify biomarkers for treatment-related neurotoxicity in patients with brain metastases using peripheral blood methylation markers previously described. The limitations of the article were appropriately recognized by the previous reviewers and the authors have made attempts to rectify the manuscript appropriately. The decision to convert to a short report and to present the study findings as hypothesis generating is appropriate, given the small sample size and other previously specified limitations.

The authors should consider clarifying the eligibility criteria for patients included in this study. The authors should also review the manuscript for clarity and English grammar and should consider limiting the extensive use of acronyms in the manuscript text. Finally, the authors should specify which cases of tumor progression or radionecrosis were biopsy confirmed, and acknowledge the limitations of interpreting findings without a biopsy.

Overall, the study represents the results of original research and has not been previously reported or published. The experiments and analyses are well described. The conclusions are hypothesis generating and should be appropriately tempered. The article is presented in an intelligible fashion, but would benefit significantly from a careful review of English grammar and clarity. The research meets appropriate ethics standards and adheres to reporting guidelines and standards for data availability.

7. PLOS authors have the option to publish the peer review history of their article (what does this mean?). If published, this will include your full peer review and any attached files.

Reviewer #1: No

Reviewer #3: No

---

## [Author Response · Author response to Decision Letter 1]

19 Sep 2023

Response to Reviewers:

Dear Reviewers,

Thank you for reviewing our manuscript. We appreciate your comments and helpful suggestions.

We revised our paper according to reviewers` suggestions. First, we revised our terminology to the term “radiation-induced brain injury” as suggested. Also, following your advice, we edited Figure 1A to 2 separate figures (1A and 1B). Importantly we also reviewed and corrected the manuscript for grammar and syntax as recommended.

These changes and others are summarized below (marked in red color below each comment). line numbers (marked in blue color) refer to the version of “revised manuscript highlighted with track changes”.

Reviewer #1: The authors have responded thoroughly to the reviews. The work remains intriguing and novel. There are a few issues that remain that need to be addressed prior to acceptance for publication.

1. Figure 1A looks quite busy and the pattern / association with RIN/RIBD is not clear and does not support the idea that increases in cfDNA tightly correlate with brain injury. However, this may be due to noise and variation inherent to the data. Perhaps two figures would be better. One panel could look only at XRT patients without RIBD (red lines), anchored by start of XRT (align all the curves with respect to clinical onset of RIBD instead of XRT start). The RIBD panel (blue lines) might be better anchored by the date of RIBD. The patterns are just quite difficult to see in the figure 1A, which looks quite noisy, and look like they argue against any real pattern, but I suspect this can be teased out using an approach something like the above.

Adding the detailed individual case courses is very helpful. Thank you for reporting these data.

Thank you for these helpful suggestions. We edited Figure 1A as recommended, by splitting it to 2 parts: 1A (to represent patients with RIBD) and 1B (to represent patients non-RBI) and highlighting clinical onset of each event. We hope it is clearer now for visualization and interpretation.

2. For patient #11 WBRT – they appeared to have a single spike in cfDNA – did their symptoms only last a few days and then resolve? Or did they persist after cfDNA levels returned to normal? This is worth a brief comment/discussion.

I will refer my response to patient #14 that indeed has only 2 time-points of cfDNA measurement instead of patient #11 (that have 9 measurement time-points). Following one day after clinical onset his neurological symptoms were stable. Unfortunately, we were not able to continue following patient #14 later, since he was transferred to other hospital for palliative care due to systemic progression of his cancer, from which he deceased 35 days later.

3. The revisiting of the terminology is appreciated. However, in the cited paper the term “Radiation-Induced Brain Injury” appears to be used far more often than “Radiation-Induced Brain Disease” (17 vs. 3 times). The term “disease” implies a more sustained, potentially permanent condition and physiologic entity, whereas the term “Radiation Inducted Brain Injury” would be perfect and corresponds nicely to what the authors are directly claiming to measure – injury of brain cells (both neurons and glia). The term “Radiation-Induced Brain Injury” should be used throughout to avoid confusion or a pejorative connotation.

We changed the term “Radiation-induced brain disease” to the term “Radiation-Induced Brain Injury” in the manuscript, as well as in the title and in all associated figures and tables.

Minor comments:

1. Table 1 is quite simplistic and these data are probably better just described in the text, separating the descriptions of the SRS vs. WBRT patients with respect to timing of RIBD events (these are very different clinical contexts).

As suggested, we omitted table 1 and described the important patient’s characteristic in the text on the results section (lines 137-140).

2. Table S1: The definition column might be more clear to just state “Acute, Early-Delayed, or Late Delayed” instead of acronyms like ARIBD, EDRIBD, LDRBID.

We corrected this column as suggested. 

3. Table S2: Can an alternative to “URP” like “PD” be use for progressive disease?

In this work we use the term “progressive disease” to reflect tumor progression, in contrast to cases of radiological progression resulting from radiation effects on the peritumoral tissue such as peritumoral edema and inflammatory response (here referee as “Radiation induced brain injury”). We used the term 

“undetermined radiological progression (URP)” in cases that we could not attribute the radiological progression specifically to either tumor enlargement (“PD”) or peritumoral enlargement (“RBI”), In order to differentiate between those 2 entities. Since we cannot say clearly that those cases defined as “URP” are related to tumor progression we would avoid the use of the term “progressive disease” in those specific undetermined cases.

4.Table S4: Does the X axis indicate time? Is this to scale? If so time should be added. The figure is helpful for spelling out the clinical course of each patient.

Thank you for this comment. Assuming this important comment refers to figure S1 (right after table S4) - We added in figure S1 a label for the X axis which indicates it relates to time.

Reviewer #3: The authors are to be applauded for seeking to identify biomarkers for treatment-related neurotoxicity in patients with brain metastases using peripheral blood methylation markers previously described. The limitations of the article were appropriately recognized by the previous reviewers and the authors have made attempts to rectify the manuscript appropriately. The decision to convert to a short report and to present the study findings as hypothesis generating is appropriate, given the small sample size and other previously specified limitations.

1.The authors should consider clarifying the eligibility criteria for patients included in this study. 

Thank you for this comment. As suggested, we elaborate eligibility criteria in the “Methods” section describing patient population (lines 72-74).

2.The authors should also review the manuscript for clarity and English grammar and should consider limiting the extensive use of acronyms in the manuscript text. 

We reviewed the manuscripts and rephrased several sentences to be clearer, as well as several corrections related to grammar and syntax. As suggested we also omitted some of the abbreviations and used instead its full terms (“brain metastases”, “early-delated”, “ late-delayed”, progressive disease”, “tumor response”, “stable disease”). ( lines: 24, 25, 28, 30, 37, 44, 55, 62, 63, 67-68, 72, 73-74, 85-87, 92, 98-104, 123, 173-176, 181, 184187, 189, 191, 193, 195, 199-200, 217, 222, 225, 228, 230, 231, 233, 236, 238, 242, 244, 248, 250, 264, 265, 270, 272, 274, 275),

3. Finally, the authors should specify which cases of tumor progression or radionecrosis were biopsy confirmed, and acknowledge the limitations of interpreting findings without a biopsy.

We added a sentence to reflect this limitation in lines 88-89, and in the discussion section, in lines 281-282.

---

## [Decision Letter · Decision Letter 2]

16 Oct 2023

Short Report: Plasma based biomarkers detect radiation induced brain injury in cancer patients treated for brain metastasis: A Pilot study.

PONE-D-23-10042R2

Dear Dr. Zick,

We’re pleased to inform you that your manuscript has been judged scientifically suitable for publication and will be formally accepted for publication once it meets all outstanding technical requirements.

Kind regards,

Ryuya Yamanaka, M.D.,Ph.D.

Academic Editor

PLOS ONE

Additional Editor Comments (optional):

Reviewers' comments:

Reviewer's Responses to Questions

**Comments to the Author**

1. If the authors have adequately addressed your comments raised in a previous round of review and you feel that this manuscript is now acceptable for publication, you may indicate that here to bypass the “Comments to the Author” section, enter your conflict of interest statement in the “Confidential to Editor” section, and submit your "Accept" recommendation.

Reviewer #1: All comments have been addressed

Reviewer #3: All comments have been addressed

2. Is the manuscript technically sound, and do the data support the conclusions?

Reviewer #1: Yes

Reviewer #3: Yes

3. Has the statistical analysis been performed appropriately and rigorously? 

Reviewer #1: Yes

Reviewer #3: Yes

4. Have the authors made all data underlying the findings in their manuscript fully available?

Reviewer #1: Yes

Reviewer #3: Yes

5. Is the manuscript presented in an intelligible fashion and written in standard English?

Reviewer #1: Yes

Reviewer #3: Yes

6. Review Comments to the Author

Reviewer #1: All comments have been addressed. The paper is improved as a result with respect to terminology and a clear presentation of all the data.

Reviewer #3: The authors have addressed our concerns.

7. PLOS authors have the option to publish the peer review history of their article (what does this mean?). If published, this will include your full peer review and any attached files.

Reviewer #1: No

Reviewer #3: No

---

## [Editor Report · Acceptance letter]

14 Nov 2023

PONE-D-23-10042R2 

Short Report: Plasma based biomarkers detect radiation induced brain injury in cancer patients treated for brain metastasis: A Pilot study. 

Dear Dr. Zick:

I'm pleased to inform you that your manuscript has been deemed suitable for publication in PLOS ONE. Congratulations! Your manuscript is now with our production department. 

Kind regards, 

on behalf of

Professor Ryuya Yamanaka 

Academic Editor

PLOS ONE